# Route Planning for Agricultural Machines with Multiple Depots: Manure Application Case Study

**Mahdi Vahdanjoo** [1],[*],[†] , **Kun Zhou** [2],[†] and **Claus Aage Grøn Sørensen** [1]

1   Department of Engineering, Faculty of Technical Sciences, Aarhus University, Finlandsgade 22,
    8200 Aarhus N, Denmark; claus.soerensen@eng.au.dk
2   Research & Advanced Engineering, AGCO A/S, 8930 Randers, Denmark; kun.zhou@agcocorp.com
*   Correspondence: mahdi.vahdanjoo@agro.au.dk
†   These authors contributed equally to this work.

**Abstract:** Capacitated field operations involve input/output material flows where there are capacity constraints in the form of a specific load that a vehicle can carry. As such, a specific normal-sized field cannot be covered in one single operation using only one load, and the vehicle needs to get serviced (i.e., refilling) from out-of-field facilities (depot). Although several algorithms have been developed to solve the routing problem of capacitated operations, these algorithms only considered one depot. The general goal of this paper is to develop a route planning tool for agricultural machines with multiple depots. The tool presented consists of two modules: the first one regards the field geometrical representation in which the field is partitioned into tracks and headland passes; the second one regards route optimization that is implemented by the metaheuristic simulated annealing (SA) algorithm. In order to validate the developed tool, a comparison between a well-known route planning approach, namely B-pattern, and the algorithm presented in this study was carried out. The results show that the proposed algorithm outperforms the B-pattern by up to 20.0% in terms of traveled nonworking distance. The applicability of the tool developed was tested in a case study with seven scenarios differing in terms of locations and number of depots. The results of this study illustrated that the location and number of depots significantly affect the total nonworking traversal distance during a field operation.

**Keywords:** operations management; optimization; route planning; field efficiency; multiple depots; simulated annealing; nonfixed MD-CVRP problem; decision support

---

## 1. Introduction

The decreasing marginal income and increasing costs in agriculture require modern agriculture to become increasingly productive [1]. Over the last decades, the production of large and powerful agricultural machinery has been the main engineering focus of manufacturing sectors to increase productivity and efficiency. Although usage of these large machines can increase the unit capacity, it also leads to undesirable and adverse environmental and biological effects such as soil compaction, which potentially reduces the yield [2,3]. Therefore, more efforts are being contributed to the development of advanced information and communication technology (ICT) systems and operation management tools to achieve higher operational efficiency and machinery productivity [4]. These tools/systems have been developed ranging from aiding and supporting navigation efforts to full autosteering systems [5–8]. Most of these systems can navigate and supervise the operator to complete the field tasks by setting the route for a vehicle either manually or using a predefined fieldwork pattern.

In recent years, a larger number of vehicle route planning algorithms have been developed to be integrated with the aforementioned systems. Commonly, the objectives of these algorithms are

twofold: one is creating a geometrical representation of the operational environment in which the field is divided into a set of parallel, straight, or curved tracks and headland passes, and several methods addressing field representation have been developed [9,10]. The other regards the optimization of the routing of vehicles within the geometrically represented field. Concerning this problem, advanced methods based on combinatorial optimization have recently been introduced based on one or multiple optimization criteria that include the minimization of the total nonworking traveled distance [11], total operational time [12], and soil compaction risk [13,14]. All of these algorithms used the metaheuristic algorithms (i.e., Clarke–Wright savings algorithm [15,16], genetic algorithm [17,18], simulated annealing algorithm [6,19], ant colony algorithm [20,21]) to find optimal or near-optimal solutions in a reasonable time. The reason is that the route planning problem is considered as a non-deterministic polynomial (NP-hard) combinatorial optimization problem, and exact algorithms tend to perform poorly in the case of large-size instances [22].

All of the aforementioned algorithms only take into account one depot as the replenishment place during field operations; however, in some operations, multiple depots might be required. For instance, in a capacitated operation such as a manure application, several refillings are needed to complete the entire operation; in a case that the slurry storage facilities are far from the field in order to reduce the total nonworking traveled distance and subsequently increase the field efficiency, farmers often set up multiple slurry buffer tanks (as depots) at different locations for refilling so that each trip can begin and end at different depots. Regarding the multiple depot vehicle route planning problem (MDVRP), it has not received much attention in the agricultural sector. While in the industry sector, this problem has been extensively studied by multiple authors [23–26]. Due to the specific nature of agricultural operations, existing methods from industry cannot be directly applied. To this end, in this paper, a route planning tool for the MDVRP problem is developed and the optimization problem is solved by using the simulated annealing algorithm. The remainder of this paper is organized as follows. In Section 2, the methodology and mathematical formulation of the problem are explained. The algorithm of the developed tool is validated by comparing it with a well-known route planning method B-pattern [27], and then its applicability is demonstrated in the manure application process considered in Section 3. Finally, in Section 4, the conclusions are drawn.

## 2. Materials and Methods

### 2.1. Problem Definition

Agricultural field operations such as seeding, spraying, and fertilization are executed by a fleet of machines with limited capacity. To achieve the optimal performance of such machines, a route with several trips back and forth between depots needs to be carefully planned. A trip is an operational cycle that consists of suboperations carried out by a machine: 1. filling the manure tank at a depot, 2. leaving that depot to process the field tracks, and 3. driving back to a depot for a new refilling. In addition, as a frequent practice of farmers and machine operators, the distributor can only turn in the headland area in order to reduce the soil compaction and avoid crop damage, and each track is treated only once.

In this study, the manure application process is selected as the study case where multiple refilling depots are considered. The routing problem in our case is formulated as a capacitated vehicle routing problem with multiple depots (MD-CVRP) that has been extensively studied in an industrial section like logistic planning. This problem can be classified into two different categories: fixed and nonfixed destination MD-CVRP. The fixed destination MD-CVRP cannot apply to our research problem, in which each trip begins and ends at the same depot. Therefore, a nonfixed destination MD-CVRP must be developed for this study.

## 2.2. Mathematical Formulation

In this study, all the fieldwork tracks are defined as a set of directed arcs, and each arc has the form of $a = (v_i, v_j)$ where $v_i, v_j$ is called a tail and head vertex, respectively. In this way, each fieldwork track is represented by two arcs with the same length but in opposite directions. These two arcs are called sibling arcs. Therefore, to cover the entire set of tracks, one of these sibling arcs should opt in the optimized solution and a depot is defined as a special arc, which starts and ends at the same point and its sibling is itself. Bochtis and Vougioukas [11] formulated this problem as a graph $G = (V, A)$, $V = V_d \cup V_t$, where $V_d = \{v_1, v_2, \ldots, v_m\}$ is the depot vertex sets and $V_t = \{v_{m+1}, v_{m+2}, \ldots, v_{m+n}\}$ is the vertex sets corresponding to the ending points of fieldwork tracks. The $V_t$ further is classified into two sets: even $V_{te}$ and odd $V_{to}$ vertices. Therefore $A_t = \{a = (v_i, v_j) : v_i, v_j \in V_{to}, V_{te}, |i - j| = 1\}$ can be considered as the set of arcs representing tracks, and $A_d = \{a = (v_i, v_i) : v_i \in V_d\}$. Therefore, the number of arcs $A = A_t \cup A_d$ is equal to $m$, which is the number of vertices. Each arc $a$ is assigned a value $y_t$, where $y_t$ is equal to the total demand of arc $a$ when $a \in A_t$, otherwise $y_t$ is $\infty$, meaning that the depot arc has an infinite capacity for the refilling machine unit. The total demand for a field can be calculated by the total summation of values of all arcs' demand divided by two. In addition, a fleet of $m$ homogeneous vehicles of capacity $Q$ is considered in the problem. Each arc $a_{ij}$ has a related traveling cost $C_{ij}$.

The nonfixed destination MD-CVRP is to consider $m$ vehicle routes with these criteria:

- Every path should start and end at the depot;
- Every track is covered exactly once;
- The total demand for any vehicle path should not exceed the capacity of the vehicle.

For any path, the total nonworking distance can be defined as $f(\partial)$, which is the summation of distances from the starting depot to the first fieldwork track in the planned track sequence and all headland turning distances, and finally the movement from the last track to the final depot. The optimal solution which minimizes the cost function is the permutation $\partial^*$ and can be calculated as follows:

$$\partial^* = argmin_{\partial} f(\partial) \tag{1}$$

The decision variable $X_{ij}$ of the formula is defined as follows:

$$X_{ij} = \{1 : (if\ the\ vehicle\ immediately\ goes\ from\ vertex\ i\ to\ vertex\ j)\ ; \\ 0 : (if\ the\ vehicles\ need\ to\ visit\ depots\ for\ refilling)\} \tag{2}$$

The objective function can be defined as follows:

$$Min : \sum_{i \in (v_t \cup v_d)} \sum_{j \in (v_t \cup v_d)} C_{ij} X_{ij} \tag{3}$$

The constraints are as follows:

$$\sum X_{ij} + X_{ji} = 1, \forall\ (i, j) \in A \tag{4}$$

$$\sum_{j \in v_t} X_{ij} = 1, \forall\ i \in V_d \tag{5}$$

$$\sum_{i \in v_t} X_{ij} = 1, \forall\ j \in V_d \tag{6}$$

$$\sum_{i, j \in A} X_{ij} Y_{ij} \leq Q, \left( Y_{ij} = demand\ arc\ from\ vertice\ i\ to\ j \right) \tag{7}$$

$$\sum_{i \in v_t} X_{ik} - \sum_{j \in v_t} X_{kj} = 0, \forall\ k \in (V_t \cup V_d) \tag{8}$$

$$\sum_{i \in H} \sum_{j \in H} X_{ij} \leq \|H\| - 1, \forall\ H \subseteq V_t, \|H\| > 1 \tag{9}$$

$$X_{ij} \in \{0, 1\}, \ \forall \ i, j \in (V_t \cup V_d) \tag{10}$$

Equation (4) states that only one of the sibling arcs needs to part of the solution. Equations (5) and (6) ensure that each route should be started and finished at a depot. Equation (7) specifies that the total summation of demand related to each trip should not exceed the capacity of the distributor's tank. Equation (8) represents that if the vehicle starts covering a track K, it will leave the track at the end of the process. Equation (9) will remove the subtours from feasible solutions. The last equation, (10), states the type of the variable.

### 2.3. Methodology

The problem explored in this study is to find the optimal traversal sequence of tracks in a field with multiple depots by considering the capacity of the distributor to minimize the total nonworking distance. Therefore, this problem can be classified as a well-known multidepot capacitated vehicle routing problem (MD-CVRP), which is an NP-hard problem and the tracks' arcs are going to visit instead of customers, and to complete the task, the distributor should visit the depots several times for refilling. A metaheuristic algorithm, simulated annealing (SA), was selected for finding the solution for this problem.

### 2.3.1. Field Representation

Field representation is a process of generating a geometrical representation of a field area to provide a concise representation of the operational environment, which includes three tasks: generation of headland passes (H), determination of the fieldwork tracks (T) that completely cover the field area, and generation of the connecting paths that connect the depots, tracks, and headland passes. Specifically, these connecting paths that demonstrated in Figure 1 are:

- A path connects fieldwork tracks' ends to adjacent fieldwork tracks' ends (T2T);
- A path connecting each depot and to the first headland pass (D2H);
- A path connecting each headland pass to its adjacent headland pass (H2H);
- A path connecting fieldwork tracks ends at each headland pass (T2H).

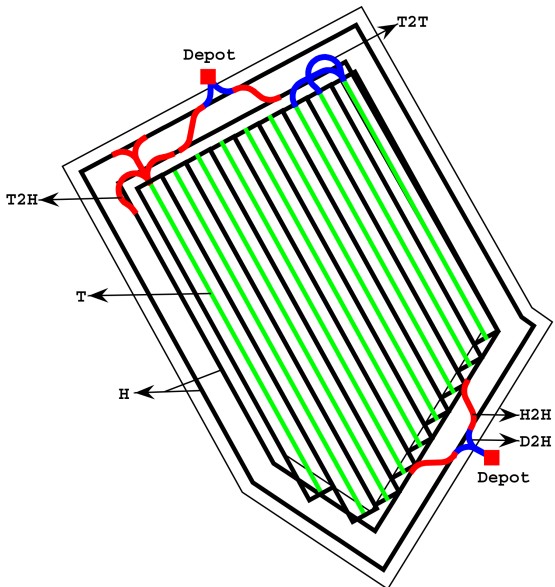

**Figure 1.** An illustrative example of field representation. H: headland passes; T: fieldwork track; T2H: fieldwork track to headland passes; T2T: fieldwork track to fieldwork track; H2H: headland passes to headland passes; D2H: depot to headland passes.

It is worth noting that the curvature of connecting paths is constrained by the minimum turning radius of the machine for making the paths traversable by machines.

### 2.3.2. Cost Matrix Generation

The cost matrix gives the transition cost between every pair of vertices of the field graph. The field graph consists of all the edges in the field with their corresponding weight (which is the actual length of the edge). The cost matrix is square ($n * n$) with an element $C_{ij}$ which represents the transition cost from vertex $i$ to vertex $j$, where $i \neq j$; otherwise, the cost is equal to 0. The shortest path search algorithm was applied for calculating the transition cost between every pair of nodes [28].

### Simulated Annealing Algorithm

Simulated annealing (SA) as a stochastic algorithm is used in this study to investigate the near-optimal solution. This method has been implemented in several problems in various fields such as computer design, route planning, and image processing [6,28,29]. The mechanism of this algorithm is based on the simulation of a cooling process called annealing where a solid is gradually cooled. The SA algorithm has some primary features such as the initial temperature, cooling rate, and termination policy, which should be defined carefully in advance since the performance of the algorithm depends on them. For instance, at the beginning when we have a higher temperature, the probability of accepting newer worse solutions is high and with time it will decrease. By this ability, the SA algorithm enables a search of all solution space and an escape from trapping in local minima [29].

All of the SA algorithm steps can be listed as follows:

1.  Define the SA main parameters such as initial temperature = 200, temperature–damping rate = 0.9.
2.  SA main loop

    a.  Generate initial solution

        I.   Generate the initial solution based on the maximum capacity of the vehicle to determine when it is needed to go to the depot for refilling
        II.  Calculate the cost of an initial solution by using the cost function that represents the total nonworking distances traveled by the vehicle

    b.  Set the temperature (T) equal to the initial temperature ($T_0$)
    c.  Update the best solution ever found

3.  SA inner loop based on T

    a.  Generate a neighborhood solution based on the initial solution

        I.   Generate the neighborhood solution based on the maximum capacity of the vehicle
        II.  Calculate the cost of neighborhood solution

    b.  Compare the cost of the new solution with the initial solution
    c.  If the new solution is convincing, then accept it as the best solution
    d.  If the new solution is not convincing, then there is an opportunity to accept it by the condition based on T
    e.  Update the best solution
    f.  Reduce the T based on the damping rate ($\alpha$).

A flowchart (Figure 2) summarizes all the steps of the SA algorithm:

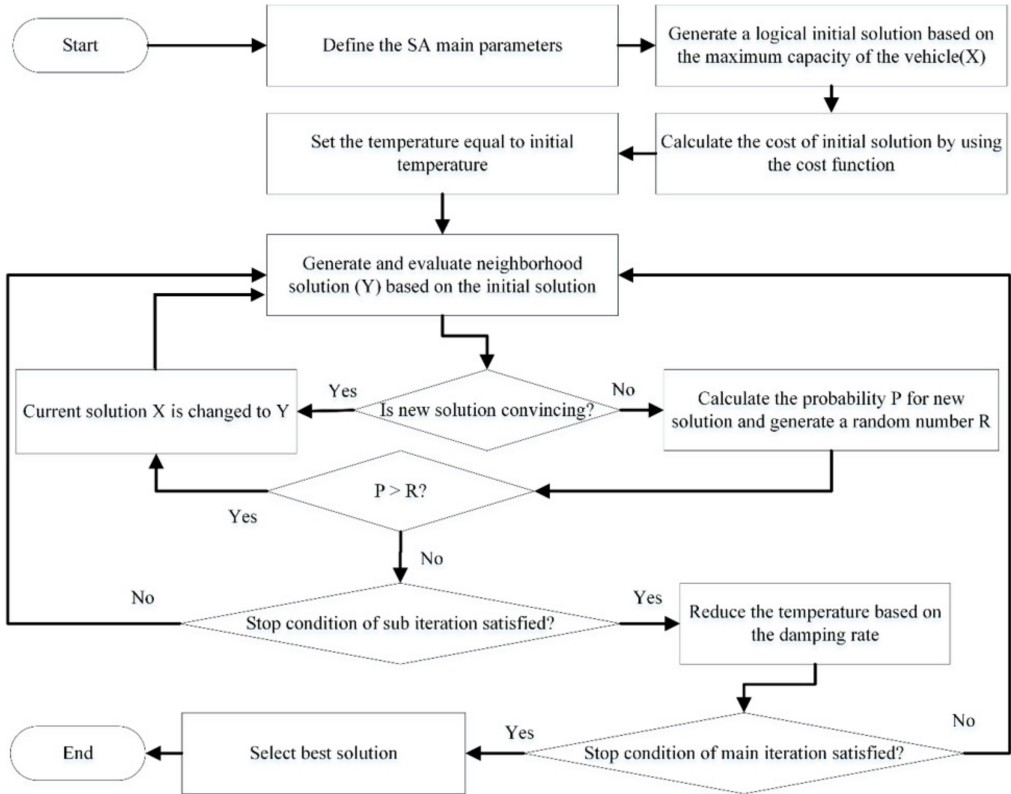

**Figure 2.** An overview of the simulated annealing algorithm.

Initial Solution Generation

Before the initial solution construction, all arcs are clustered based on the distance of its head to its nearest depot. For example, as presented in Figure 3, the nearest depot to arc 3 is depot 1, while arc 4 is depot 2. The construction criteria are:

- Each solution should be initiated and ended with a depot that can be the same or different.
- For simplicity, only one of each sibling arcs is presented in the solution.
- Based on the solution already constructed, if the remaining capacity of the agricultural machine is not sufficient to process the next track, then the nearest depot has to be selected for refilling service.

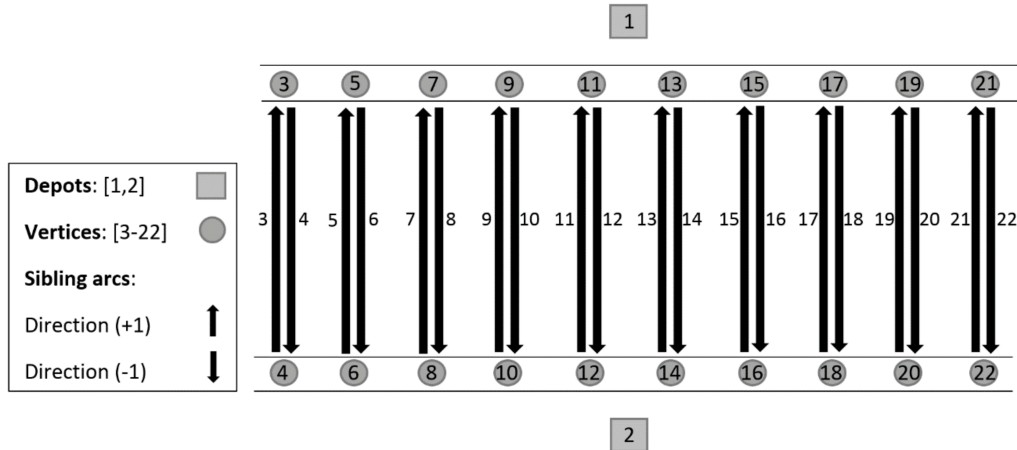

**Figure 3.** An example of arc representation for a field with 2 depots and 10 tracks.

The steps of creating an initial solution are summarized in a flowchart (Figure 4):

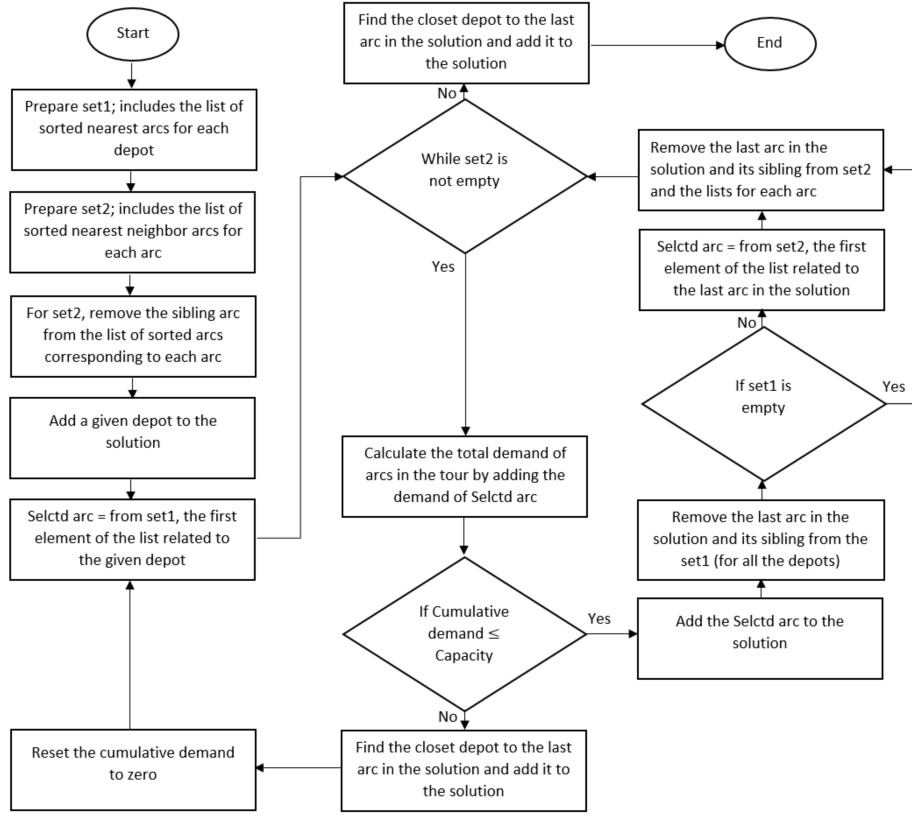

**Figure 4.** An overview of the process of generating an initial solution.

Solution Evaluation

In order to evaluate the quality of a solution, a cost function is defined, specifically for the traveling cost. Let $C(a_i)$ be the traveling cost for covering the arc $a_i$. Then for a solution, the total traversal cost of the agricultural vehicle is equal to the summation of the costs $C(a_i)$ corresponding to the arc $a_i$, which is traversed by the vehicle. The total cost function $F(a)$ can be defined as follows: $F(a) = \sum_{i \in (\textit{visited arcs list})} C(a_i)$.

Neighborhood Solution Generation

Random swaps, random swaps of subsequences, random insertions, random insertions of subsequences, reversing a subsequence, random swaps of reversed subsequences, and finally random insertions of reversed subsequences are among the neighborhood operators that can be used for making a new solution in the SA algorithm. Table 1 shows how these operators make changes in the initial solution.

**Table 1.** Applied neighborhood operators [6].

| | | | | | | | | | | | | | | | | | | | | | | | | | |
|---|---|---|---|---|---|---|---|---|---|---|---|---|---|---|---|---|---|---|---|---|---|---|---|---|---|
| | **Random swaps** | | | | | | | | | | | | | **Random swaps of subsequences** | | | | | | | | | | | |
| B * | 0 | 9 | 4 | 7 | 2 | 5 | 12 | 0 | **13** | **18** | 19 | 16 | 0 | 0 | 9 | 4 | 7 | 2 | 5 | **12** | 0 | 13 | 18 | 19 | 16 | 0 |
| A * | 0 | 9 | 4 | 7 | 2 | 5 | 12 | 0 | **18** | **13** | 19 | 16 | 0 | 0 | **5** | **12** | **2** | **9** | **4** | **7** | 0 | 13 | 18 | 19 | 16 | 0 |
| | **Random insertions** | | | | | | | | | | | | | **Random insertions of subsequences** | | | | | | | | | | | |
| B | 0 | 9 | 4 | 7 | 2 | **5** | 12 | 0 | 13 | 18 | 19 | 16 | 0 | 0 | 9 | 4 | 7 | **2** | **5** | **12** | 0 | 13 | 18 | 19 | 16 | 0 |
| A | 0 | 9 | 4 | **5** | 7 | 2 | 12 | 0 | 13 | 18 | 19 | 16 | 0 | 0 | 9 | **2** | **5** | **12** | **4** | 7 | 0 | 13 | 18 | 19 | 16 | 0 |
| | **Reversing a subsequence** | | | | | | | | | | | | | **Random swaps of reversed subsequences** | | | | | | | | | | | |
| B | 0 | 9 | **4** | **7** | **2** | **5** | **12** | 0 | 13 | 18 | 19 | 16 | 0 | 0 | 9 | **4** | **7** | 2 | 5 | **12** | 0 | 13 | 18 | 19 | 16 | 0 |
| A | 0 | 9 | **12** | **5** | **2** | **7** | **4** | 0 | 13 | 18 | 19 | 16 | 0 | 0 | **12** | **5** | 2 | **7** | **4** | **9** | 0 | 13 | 18 | 19 | 16 | 0 |

\* B = Before, A = After. Numbers with bold changed after applying the neighborhood operator.

## 3. Results and Discussion

The SA algorithm was implemented in MATLAB® technical programming language (The MathWorks Inc., Natwick, MA, USA) on a computer with the following specifications: Intel ® Core ™ i5-3210 M CPU @ 2.50GHZ with 4GB RAM. To investigate the quality of the solutions achieved by the algorithm developed, the results presented by the well-known approach (B-pattern) [11] are used for comparison. Furthermore, [29] also compared their proposed algorithm with the B-pattern approach. The field shapes and machine-related specifications were considered in this work as shown in Table 2, which are used as input data for the optimization algorithm. The capacity of the machine is set to infinite because disk-harrowing was considered as the field operation in [11].

**Table 2.** Input data related to the fields and applied machine.

| Field | Field Size | Operating Width (m) | Minimum Turning Radius (m) |
|:---:|:---:|:---:|:---:|
| 1 | 24 m × 30 m | 2.89 | 3.5 |
| 2 | 50 m × 80 m | 2.5 | 3 |
| 3 | 30 m × 40 m | 2.5 | 3 |
| 4 | 30 m × 70 m | 2.5 | 3 |

In the B-pattern approach [11], the authors apply their proposed method in four fields. Field numbers 1 to 4 are divided into 8, 20, 12, and 12 tracks, based on the defined operating width of the machine and the size of those fields. The results achieved from the comparison of the proposed optimization algorithm in this study with the solutions provided in [11] and [29] are presented in Table 3. The comparisons illustrate that all the results from the algorithm presented in this study outperformed the solutions provided by [11]. The total nonworking traveled distance in the headland part for the first field, based on the presented method in this study, is 94.439 m, which shows a 1.4% improvement in comparison with the quality of the solution found in [11]. Moreover, the total nonworking traveled distance in the headland area for field numbers 2 to 4, based on the proposed approach in this work, is 232.566, 141.027, and 141.027 m, respectively. The comparison of the results achieved in this work with the solutions presented in [11] shows that our solutions are 1.4%, 20%, and 15.3% more efficient than those presented in [11]. Moreover, the comparison shows that the solutions presented in this work are 1.2%, 3.4%, and 3.1% more efficient than those provided in [29] for field numbers 2, 3, and 4, respectively.

**Table 3.** Comparison of the generated solutions based on the proposed method in this study with the results provided in studies [11] and [29]. (**A**) Method proposed in [11], (**B**) method proposed in [29], and (**C**) method proposed in this study.

| Proposed Approach by | Field | Solution (Track Order) | Headland Nonworking Distance (m) | Total Traversal Distance (m) |
|:---:|:---:|:---:|:---:|:---:|
| A ([11]) | 1 | <1,4,7,3,6,8,5,2> | 95.77 | 335.77 |
| | 2 | <20,17,14,11,8,12,9,3,6,2,5,1,4,7,10,13,16,19,15,18> | 235.92 | 1835.92 |
| | 3 | <1,5,11,7,3,9,2,6,10,4,8,12> | 176.45 | 656.45 |
| | 4 | <2,6,10,4,8,12,9,3,7,11,5,1> | 166.45 | 1006.45 |
| B ([29]) | 1 | <1,4,7,3,6,2,5,8> | 94.439 | 334.439 |
| | 2 | <20,17,14,11,8,5,2,6,3,1,4,7,10,13,9,12,16,19,15,18> | 235.491 | 1835.491 |
| | 3 | <1,4,10,7,3,6,2,5,8,11,9,12> | 146.027 | 626.027 |
| | 4 | <2,5,3,6,9,12,10,7,11,8,4,1> | 145.602 | 985.602 |
| C (This study) | 1 | <1,4,7,3,6,2,5,8> | 94.439 | 334.439 |
| | 2 | <19,16,13,10,7,4,1,3,6,9,12,15,18,20,17,14,11,8,5,2> | 232.566 | 1832.566 |
| | 3 | <2,5,1,4,8,11,7,10,12,9,6,3> | 141.027 | 621.027 |
| | 4 | <2,5,1,4,7,10,12,9,6,3,8,11> | 141.027 | 981.027 |

Figure 5 shows the coverage of all four mentioned fields with the type of turnings used for traveling from one track to another.

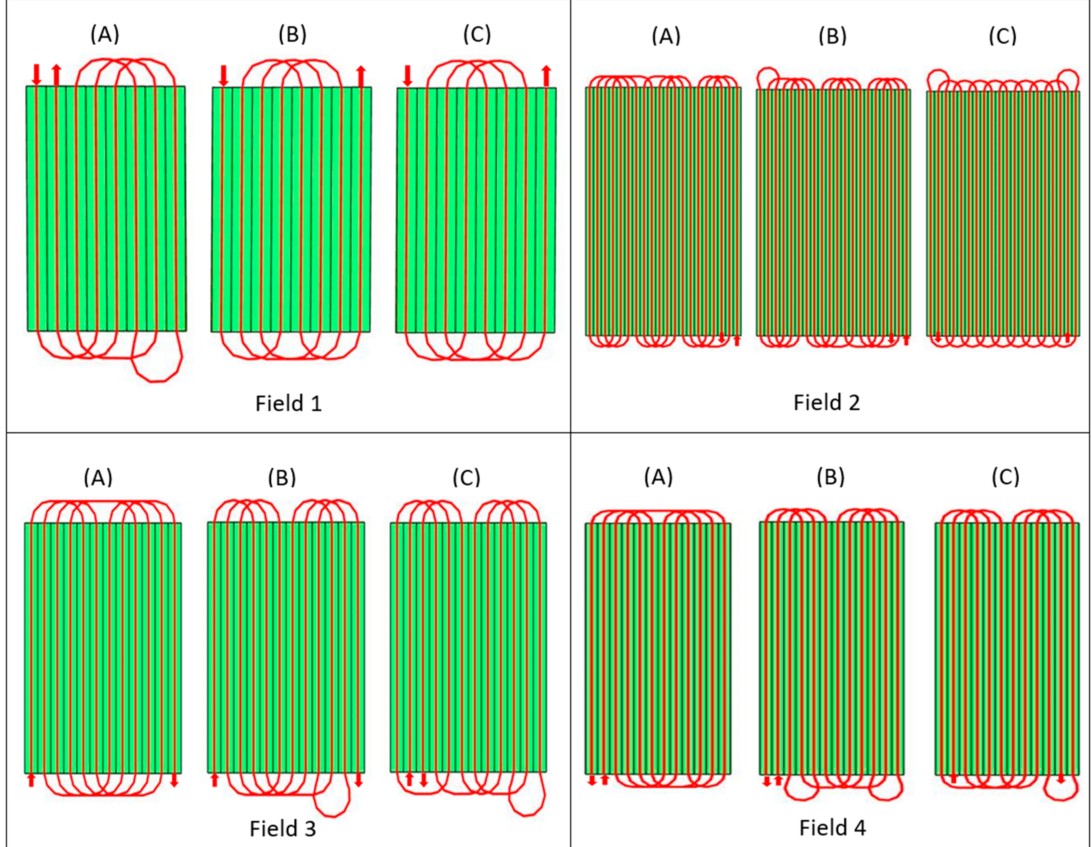

**Figure 5.** Comparison of the results obtained by applying different approaches: (**A**) method proposed in [11], (**B**) method proposed in [29], and (**C**) method proposed in this study. The track's index starts from the left and moves to the right in each field.

To demonstrate the applicability of this method for the multiple-depot route planning problem, seven scenarios for the manure application process were carried out based on the number of the depots and their locations in the field, as well as the distributor specifications. Specifically, the tested field is selected from the GIS database of the Danish Ministry of Food, Agriculture, and Fisheries, and the information regarding the properties of the field is presented in Table 4.

**Table 4.** Properties of the case study field (size = 16 hectares).

| Properties | Latitude | Longitude |
| --- | --- | --- |
| Sample field | 56°36′17.21′′ N | 10°13′41.68′′ E |
| Depot 1 | 56°36′4.6074′′ N | 10°13′37.777′′ E |
| Depot 2 | 56°36′11.0982′′ N | 10°13′34.1601′′ E |
| Depot 3 | 56°36′25.046′′ N | 10°13′45.1398′′ E |

Figure 6 illustrates the location of possible depots and field partitioning. The technical characteristics of the distributor are presented in Table 5. In this regard, the field can be divided into 25 fieldwork rows. The total demand of the field is equal to 4235 (liter), therefore at least five refilling trips between the field and depot are needed to cover the entire field. A total of seven scenarios are considered where they only differ in the locations and number of depots used.

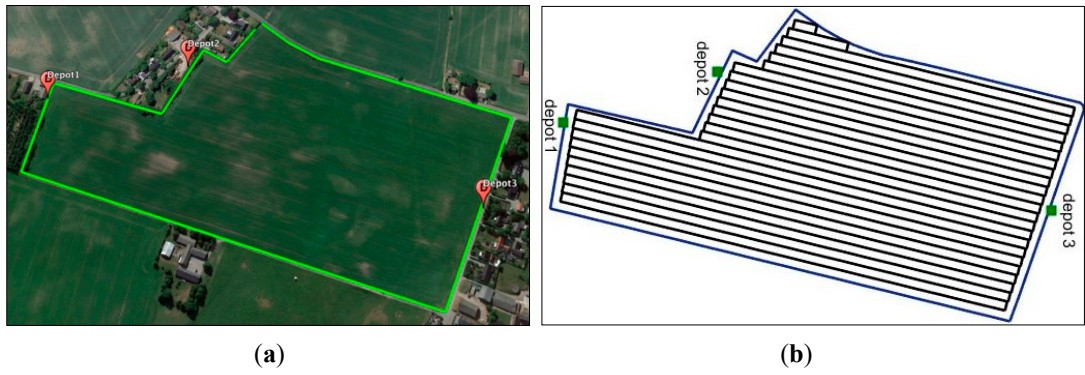

|  |  |
|:---:|:---:|
| (**a**) | (**b**) |

**Figure 6.** (**a**) Field representation for all the scenarios and (**b**) field partitioning. The arrow shows the order of track Id in the sample field.

**Table 5.** Technical characteristics of the applied manure distributor.

| Technical Characteristics of the Distributor | Values |
|:---:|:---:|
| Capacity (m$^3$) | 25 |
| Minimum turning radius (meter) | 9 |
| Operating width (meter) | 12 |
| Application rate (m$^3$/ha) | 17 |

Table 6 shows the list of tracks generated after partitioning the sample field. Each track has a pair of sibling arcs and in the provided solution for simplicity, only one of these sibling arcs is demonstrated. For instance, arc 1 starts from node 1 and ends at node 2 and arc 2 is from node 2 to node 1.

**Table 6.** The list of track Id and corresponding sibling arcs and demand (liter).

| Track Id | 1 | 2 | 3 | 4 | 5 | 6 | 7 | 8 | 9 |
|:---:|:---:|:---:|:---:|:---:|:---:|:---:|:---:|:---:|:---:|
| Arcs | (1,2) | (3,4) | (5,6) | (7,8) | (9,10) | (11,12) | (13,14) | (15,16) | (17,18) |
| Demand | 12,157 | 12,205 | 12,252 | 12,300 | 12,348 | 12,395 | 12,443 | 12,491 | 12,538 |

| Track Id | 10 | 11 | 12 | 13 | 14 | 15 | 16 | 17 | 18 |
|:---:|:---:|:---:|:---:|:---:|:---:|:---:|:---:|:---:|:---:|
| Arcs | (19,20) | (21,22) | (23,24) | (25,26) | (27,28) | (29,30) | (31,32) | (33,34) | (35,36) |
| Demand | 12,583 | 9795 | 9186 | 9151 | 9116 | 9080 | 9045 | 9009 | 8974 |

| Track Id | 19 | 20 | 21 | 22 | 23 | 24 | 25 |
|:---:|:---:|:---:|:---:|:---:|:---:|:---:|:---:|
| Arcs | (37,38) | (39,40) | (41,42) | (43,44) | (45,46) | (47,48) | (49,50) |
| Demand | 8938 | 8646 | 8116 | 8038 | 7956 | 2047 | 629 |

Table 7 illustrates all the scenarios with their corresponding solution, including the optimal order of arcs to cover. To better compare the defined scenarios, the total traversal nonworking distance in the field is calculated.

The results in Table 7 show that the locations of the depots affect the nonworking traversal distance in the field. By comparing the three scenarios (1, 2, 3) in which only one depot is considered, it can be seen that scenario 3 produces the best solution. Scenario 3 with 3220.9 (m) nonworking traversal distance has 35.6% and 40% cost savings in comparison with scenarios 1 and 2, respectively. Furthermore, a different combination of depots in scenarios 4 to 6 illustrates how the position and the order of visiting depots can affect the traversal distance. As an example, scenario 6 shows approximately 8% cost saving in comparison with scenario 4.

**Table 7.** Comparison of scenarios for the case study field.

| Scenario | Depot ID | Solution | Nonworking Distance (m) | Run Time (s) |
|---|---|---|---|---|
| 1 | (d1) | <d1, 22, 17, d1, 16, 11, d1, 32, 33, d1, 48, 49, 37, d1, 42, 46, 39, d1, 20, 23, d1, 30, 35, d1, 8, 3, d1, 10, 13, d1, 6, 1, d1, 28, 25, d1> | 5002.9 | 104 |
| 2 | (d2) | <d2, 40, 35, d2, 22, 13, d2, 44, 47, 49, d2, 8, 3, d2, 2, 5, d2, 42, 45, d2, 26, 31, d2, 34, 37, d2, 20, 23, d2, 30, 27, d2, 12, 15, d2, 18, 9, d2> | 5373 | 94 |
| 3 | (d3) | <d3, 9, 20, d3, 31, 36, d3, 21, 26, d3, 11, 16, d3, 13, 18, d3, 45, 50, 48, d3, 41, 38, d3, 1, 6, d3, 27, 24, d3, 3, 8, d3, 33, 30, d3, 39, 44, d3> | 3220.9 | 93 |
| 4 | (d1&d2) | <d1, 9, 2, d1, 5, 14, d1, 7, 16, d1, 17, 24, d2, 37, 32, d2, 27, 22, d2, 35, 30, d2, 49, 47, 44, d2, 45, 40, d2, 41, 34, d2, 25, 20, d1, 3, 12, d1> | 2978.4 | 130 |
| 5 | (d1&d3) | <d1, 1, 10, d1, 3, 12, d1, 7, 6, d1, 19, d3, 38, 29, d3, 18, 21, d3, 40, 41, d3, 24, 31, d3, 28, 25, d3, 36, 33, d3, 44, 49, 48, 45, d3, 14, 15, d3> | 2830 | 134 |
| 6 | (d2&d3) | <d2, 35, 30, d2, 37, 46, 43, d3, 22, 23, d3, 16,7, d3, 28, 25, d3, 12, 19, d3, 14, 5, d3, 18, 9, d3, 40, 47, 50, 41, d3, 2, 3, d3, 32, 33, d3 > | 2745.8 | 133 |
| 7 | (d1&d2&d3) | <d1, 9, 2, d1, 3, 12, d1, 7, 6, d1, 21, 30, d2, 37, 32, d2, 33, 42, d2, 35, 44, d2, 39, 46, 47, 50, d2, 27, 20, d1, 17, d3, 26, 23, d3, 14, 15, d3> | 2638.3 | 139 |

The scenario evaluation indicates that the number of depots can also influence the traversal distance. For instance, scenario 6 with a nonworking distance of 2745.8 (m) indicates 48.9% cost saving in comparison with scenario 2. Moreover, scenario 7 with three depots shows 11.4% and 50.9% cost saving in comparison with scenarios 4 and 2, respectively. Furthermore, the comparison of running time for all the scenarios shows that by increasing the number of depots, the complexity of the problem will increase and as a result, the algorithm takes more time for calculation.

The number of required depots for a field can be determined as follows. First, we need to know the rental/buying price of one slurry buffer tank (depot). Second, the amount of fuel consumption of the vehicle is required. The cost of fuel consumption for the nonworking distance traveled in each scenario can be calculated by multiplying the cost of fuel by the fuel consumption for nonworking traveled distance. The total cost for each scenario is the summation of fuel consumption cost and the price of slurry buffer tanks (depots). The location and the number of depots can affect the nonworking traveled distance and the total cost related to each scenario. The comparison of the total cost for each scenario can show the best scenario.

Since the proposed tool can reduce the in-field traveled distance and traffic intensity, it potentially can reduce the soil compaction risk in the field. Moreover, other factors also influence soil compaction, such as increasing soil organic matter [30,31], adequate crop rotation [2,32], types of soils [33], and contact pressure [14], which should be considered in assessing the soil compaction risk in the field.

Generally speaking, the application of the route planning tools in farm operations is beneficial currently due to their potential ability to reduce the cost units and increase the operational efficiency in the field. As one advantage of the tool presented in this study, restricting all the nonworking traversal distance to the headlands area makes this tool able to control the traffic in the main cropping area of the field.

The novel ICT systems in agricultural fleet management [4,34] can facilitate the implementation of this tool for navigating the vehicles in the field. Especially, the applicability of the proposed tool depends on the integration of in-field navigation technology and navigation aid systems. The optimized sequence of tracks generated by this tool can be used to navigate an agricultural vehicle in the field. In this regard, the operator of the machine can follow the turning paths generated by the field representation module in the headland area.

In the vehicle routing problem, the optimization is accomplished based on the cost matrix. In this paper, the nonworking traversal distance was used to generate the cost matrix. Potentially, the tool presented could be implemented in the fields with obstacles. Several field representation algorithms

were developed by other researchers [9,10], which considered obstacles in the field. In such tools, the cost matrix is generated based on the field representation module, and therefore the occurrence of obstacles in the field could simply be reflected in the cost matrix. This approach could also apply to fields with slopes by applying a 3D field representation algorithm [35]. To implement this tool in MVRP (multiple vehicle route planning) problems when heterogeneous vehicles with different capacity or maneuver capabilities are present, an individual cost matrix should be considered for each vehicle.

For future studies, the tool provided can be modified to handle other types of VRP problems. For instance, in the SVRP (stochastic vehicle route planning) problem, the proposed tool can reflect the uncertainty by stochastic costs in the cost matrix [36,37]. Furthermore, the real-time calculation of deviations between the expected and real progress of route planning can be added to this tool, which is useful for illustrating the precision of the applied tool.

## 4. Conclusions

In this paper, a tool for optimal route planning for agricultural machines with multiple depots was developed. The optimization model developed was validated by comparing it to the well-known route planning approach, B-pattern. The comparison illustrates that the proposed algorithm outperformed the B-pattern approach up to 20% in terms of reducing the nonworking traveled distance in the field. Moreover, the applicability of the presented tool for multiple depots vehicle routing problem was tested in a case study. Specifically, seven scenarios were defined to consider the different combinations of multiple depots to study the effects of the quantity and the location of the depots on the nonworking traversal distances in the field. The scenario comparisons demonstrate that changing the location of the depots could make up to 40% and increasing the number of depots could bring up to 50.9% reduction in the traveled nonworking distance in the field. The tool developed can be used as part of a decision support tool to provide optimized route planning for field operations by considering field features and machinery specifications. Since this tool can generate curved paths in the field representation module, it can be used for autonomous agricultural vehicles as well as robotics for complete field coverage.

**Author Contributions:** Conceptualization, M.V. and K.Z.; methodology, M.V. and K.Z.; software, M.V. and K.Z.; validation, M.V. and K.Z.; formal analysis, M.V. and K.Z.; investigation, M.V. and K.Z.; resources, C.A.G.S.; data curation, M.V. and K.Z.; writing—original draft preparation, M.V.; writing—review and editing, M.V., K.Z. and C.A.G.S.; visualization, M.V. and K.Z.; supervision, C.A.G.S.; project administration, C.A.G.S.; funding acquisition, C.A.G.S. All authors have read and agreed to the published version of the manuscript.

**Funding:** This research received no external funding.

**Conflicts of Interest:** The authors declare no conflict of interest.

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
