# Peer review of "Route Planning for Agricultural Machines with Multiple Depots: Manure Application Case Study"

_agronomy, doi:10.3390/agronomy10101608_

Round 1
Reviewer 1 Report
The topics of trained field operations that involve material flows are an important area in research.
- It would be nice to set a general goal for the work.
- It would be good to explain a little more why the MD.CVRP standard cannot be applied directly see Line 85
- It is necessary for point 2.2. Mathematical formulation, to include formulae references and verify that equations 1-9 are correct.
- It would be good to include a discussion of the results obtained in the investigation and compare results with other similar investigations.
- It is better to provide an up-to-date literature review. There are excellent current examples published on the subject of Agronomy Journal.
Reviewer 2 Report
In this article (Route Planning For Agricultural Machines with Multiple Depots. Case: Manure Application), a route planning tool for agricultural machines with multiple depots was developed (out-of-field facilities (depot).). A meta-heuristic algorithm, Simulated Annealing (SA), was selected for finding the solution for this problem.
The authors have illustrated the general principles of the algorithm in Figure 2.
In Figure 3.4 they presented an algorithm of their own methodology.
Figures 3, 4 are a valuable contribution describing the route optimization methodology in an understandable way.
The SA algorithm was implemented in MATLAB® technical programming language (The 200 MathWorks Inc., Natwick, MA).
The innovation proposed in the presented algorithm is to establish a route with several out-of-field facilities (depots).
The presented methodology is properly described.
The algorithm of the developed tool is validated by comparing it with a well-known route planning method B-pattern. The proposed algorithm has obtained better results.
The authors performed a thorough review of the literature, the article quotes 40 items. Many of the items are the latest - 13 items from 2020, 7 items from 2019.
The layout of the article is correct. Results and discussion refer to existing scientific achievements.
- However, I believe that the authors should supplement the article with considerations related to the location and costs of several depots (out-of-field facilities ). There is no reflection on this subject. There is little information in Figure 6. Depot 3 is located far away from depots 1 and 2.
- The authors noted that farmers often set up multiple depots at different locations to replenish the fertilizer, especially when the farmland is large. (line 56-57- „All aforementioned algorithms only take into account one depot as the replenishment place during field operations, however, in some operations, multiple depots might be required, particularly, in the case of large-sized field.”). When the authors consider the fields to be large and the proposed tool can be used. The proposed test field is 16 hectares and 3 depots.
Reviewer 3 Report
The paper deals with vehicle routing problem with multiple depots. The problem is applied to routing in agriculture. The simulated annealing algorithm is used.
Notes:
- line 61 - maybe typo "that h each"
- equation 4 - very confusing, maybe replaced by X_{ij} + X_{ji} = 1 \forall (i, j) \in ....
- please provide a better explanation of why the classical logistic problem is not the same. (maybe with some example)
- Mathematical formulation: please define the Y, equation 7 contains this variable but there is no explanation. Generally, all "letters" should be defined.
- Figure 2 - may be a missing arrow from the box "Current solution X is changed to Y" to box " Generate and evaluate ..."
- I think that a better explanation of each algorithm step will be useful. The description of the algorithm is on a general level. The description on the lower level (focused on solution problem in agriculture) will be useful for the reader that wants to implement the developed algorithm in practice. We write articles for the people!
- Some discussion about algorithm performance (runtime) would be interesting.
